# RhoGAP domain-containing fusions and *PPAPDC1A* fusions are recurrent and prognostic in diffuse gastric cancer

Hanna Yang [1], Dongwan Hong [1], Soo Young Cho[1], Young Soo Park[2], Woo Ri Ko[1], Ju Hee Kim[1], Hoon Hur[3], Jongkeun Lee[1], Su-Jin Kim[4], Sun Young Kwon[5], Jae-Hyuk Lee[6], Do Youn Park[7], Kyu Sang Song[8], Heekyung Chang[9], Min-Hee Ryu[10], Kye Soo Cho[1], Jeong Won Kang[1], Myeong-Cherl Kook [1], Nina Thiessen[11], An He[11], Andy Mungall[11], Sang-Uk Han[3] & Hark Kyun Kim[1,12]

We conducted an RNA sequencing study to identify novel gene fusions in 80 discovery dataset tumors collected from young patients with diffuse gastric cancer (DGC). Twenty-five in-frame fusions are associated with DGC, three of which (*CLDN18-ARHGAP26*, *CTNND1-ARHGAP26*, and *ANXA2-MYO9A*) are recurrent in 384 DGCs based on RT-PCR. All three fusions contain a RhoGAP domain in their 3′ partner genes. Patients with one of these three fusions have a significantly worse prognosis than those without. Ectopic expression of *CLDN18-ARHGAP26* promotes the migration and invasion capacities of DGC cells. Parallel targeted RNA sequencing analysis additionally identifies *TACC2-PPAPDC1A* as a recurrent and poor prognostic in-frame fusion. Overall, *PPAPDC1A* fusions and in-frame fusions containing a RhoGAP domain clearly define the aggressive subset (7.5%) of DGCs, and their prognostic impact is greater than, and independent of, chromosomal instability and *CDH1* mutations. Our study may provide novel genomic insights guiding future strategies for managing DGCs.

[1] National Cancer Center, Goyang, Gyeonggi 10408, Republic of Korea. [2] Department of Pathology, Asan Medical Center, University of Ulsan College of Medicine, Seoul, Republic of Korea. [3] Department of Surgery, Ajou University School of Medicine, Suwon 443-380, Republic of Korea. [4] Department of Pathology, Dong-A University College of Medicine, Busan 602-812, Republic of Korea. [5] Department of Pathology, Keimyung University School of Medicine, Daegu 41931, Republic of Korea. [6] Department of Pathology, Chonnam National University Medical School, Gwangju 501-746, Republic of Korea. [7] Department of Pathology and BioMedical Research Institute, Pusan National University Hospital and Pusan National University School of Medicine, Busan 602-739, Republic of Korea. [8] Department of Pathology, School of Medicine, Chungnam National University, Daejeon 301-747, Republic of Korea. [9] Department of Pathology, Kosin University College of Medicine, Busan 49267, Republic of Korea. [10] Department of Oncology, Asan Medical Center, University of Ulsan College of Medicine, Seoul 138-736, Republic of Korea. [11] British Columbia Cancer Agency, Vancouver, BC V5Z 1L3, Canada. [12] National Cancer Center Graduate School of Cancer Science and Policy, Goyang, Gyeonggi 10408, Republic of Korea. These authors contributed equally: Hanna Yang, Dongwan Hong and Soo Young Cho.  Correspondence and requests for materials should be addressed to H.K.K. (email: hkim@ncc.re.kr) or to S.-U.H. (email: hansu@ajou.ac.kr)

Gastric cancer presents in two major histological subtypes, intestinal and diffuse-type gastric cancers (DGCs). Despite the relatively high incidence of DGC[1,2], few whole transcriptomic analyses have been performed for this histological subtype. We therefore conducted an RNA sequencing study to search for novel driver fusions in DGC.

Several fusions have previously been reported to drive gastric cancer[3–8], but few of these have been validated by subsequent studies[3,9]. The Cancer Genome Atlas (TCGA) Research Network discovered that a cryptic splice site within exon 5 of CLDN18 activates the ARHGAP26 or ARHGAP6 splice acceptor, leading to the expression of CLDN18-ARHGAP fusion transcripts in gastric cancer, especially in the genomically stable (GS) tumors[3]. The CLDN18-ARHGAP fusions retain the transmembrane domains of CLDN18 and the Rho GTPase activating protein (RhoGAP) domain of ARHGAP26/6[3]. Yun et al. found the PPP1R1B-STARD3 read-through fusion in 21.3% of gastric cancer tissues[5]. Palanisamy et al. reported that a gastric cancer expresses the AGTRAP-BRAF fusion containing the C-terminal kinase domain of BRAF (7q34) fused to the N-terminal angiotensin II type 1 receptor-associated domain of AGTRAP (1p36)[6]. The CD44-SLC1A2 fusion, which results from 11p13–15 chromosomal inversion, is found in 1–2% of gastric cancer[7]. The SLC34A2 (4p15)-ROS1 (6q22) fusion is present in 0.4% of gastric cancer and is associated with ROS1 protein overexpression[8]. Except for the CLDN18-ARHGAP fusions[9], these in-frame gene fusions have not been validated by subsequent gastric cancer publications.

Younger cancer patients express certain gene fusions at a higher frequency than older patients, including ALK or RET fusions in lung adenocarcinomas[10,11], RET/PTC1 and RET/PTC3 fusions in papillary thyroid cancer[12], EWSR1/FUS-ATF1 fusions in mesothelioma[13], and DUX4 fusions in B cell acute lymphoblastic leukemia[14]. Despite a trend for the prevalence of gene fusions in young cancer patients, no studies have systematically investigated novel fusions in young patients with DGCs due to the relative rarity of early-onset gastric cancer, which is notable for its strong enrichment of diffuse histology. We previously published a whole exome sequencing study demonstrating that CDH1 mutations are highly prevalent in early-onset DGCs[15]. CDH1 and RHOA mutations underlie unique phenotypes of DGC, such as poorly cohesive growth, but there is a subset of DGCs that are wild-type for CDH1 and RHOA[15].

## Results

**RNA sequencing of an early-onset DGC discovery set.** To identify somatic alterations in transcriptomic profiles in early-onset DGC, we performed RNA sequencing on DGCs collected from 80 young (≤ 45 years) Korean patients who had not been treated with chemotherapy or radiation[15]. The median age of this population was 38 years (range, 20–45) and 58.7% was female, as previously reported[15]. When the sequencing data were aligned using the Burrows-Wheel Aligner (BWA) to the human reference genome, hg19, median coverage of exons was 104 × [interquartile range, 91–132] and the median number of genes with ≥ 10 × coverage was 15,960 [interquartile range, 15,401–16,600]. The median total exon coverage and 5'/3' coverage ratio were 91% [interquartile range, 90%–92%] and 0.75 [interquartile range, 0.68–0.82], respectively. Microsatellite unstable tumors (MSI) have a strong immune gene expression signature and favorably respond to anti-PD-1 therapy, but the MSI tumors are relatively rare in DGC, especially in early-onset DGC[15]. Hierarchical clustering analyses of the Reads Per Kilobase Million (RPKM) data of our discovery dataset revealed four distinct clusters. One key cluster characterized by overexpression of immune-related genes included all MSI (n = 2) and Epstein-Barr virus (EBV)-

positive tumors (n = 7; Supplementary Fig. 1 and Supplementary Tables 3 and 4). Thus, a distinct cluster expressing a strong immune gene signature existed even in early-onset DGCs. We also performed RNA sequencing on 65 samples of normal tissue adjacent to the 80 tumors that had RNA sequencing data.

We applied bioinformatics algorithms such as PRADA and Trans-ABySS to the RNA-sequencing dataset to predict novel in-frame fusions, and performed RT-PCR to validate the expression of these in-frame fusion candidates (Supplementary Table 5). Twenty-five in-frame fusions were confirmed in 20 tumors from our early-onset DGC population (Table 1). Twenty-four of these in-frame fusions were novel to gastric cancer. Only the CLDN18-ARHGAP26 fusion had previously been associated with gastric cancer by The Cancer Genome Atlas (TCGA) project[3]. Notably, one of the novel in-frame fusions, EML4-ALK, was clinically-actionable but had not been previously associated with gastric cancer[16]. A tumor containing the EML4-ALK fusion had the highest ALK expression (Fig. 1a).

Of the novel in-frame fusions listed in Table 1, the following had not been previously identified in any tumor type: CTNND1-ARHGAP26, TKT-RHOA, ARFGAP2-SLC1A2[7], IFFO2-UBR4[17], PGAP3-VMP1[17], UBE2L3[18]-MAPK1, ZNF292-PREX1[19], TACC2-PPAPDC1A[20], ANXA2-MYO9A, ECT2-FABP6, LONP1-SAFB, LUC7L3-C10orf76, CLSTN1-EFCAB7, TERF2-CDH3, GTF2I-FBF1, ARMC7-PEX14, INTS12-TBCK, ELK3-NTN4, EIF4G2-UPK2, RICTOR-GHR, RNASEH2C-CFL1, ARHGAP26-NDFIP1, and IDUA-GAK (Fig. 1b). The first eight fusions, which had ARHGAP26, RHOA, SLC1A2, UBR4, PGAP3, UBE2L3, PREX1 and PPAPDC1A as fusion partners, were previously reported with different fusion partners by the gastric cancer TCGA project (CLDN18-ARHGAP26, PRKAR2A-RHOA, CD44-SLC1A2, PIK3CD-UBR4, STARD3-PGAP3, and HNF1B-PGAP3) or in the literature[7,17–21]. These findings suggest that the dysregulation of the identified genes may have functional roles in gastric cancer.

| Gene name | | mRNA Breakpoint | |
|-----------|---------|------------------|------------------|
| 5' gene | 3' gene | 5' gene | 3' gene |
| CLDN18 | ARHGAP26 | g.chr3:137,749,947 | g.chr5:142,393,645 |
| CTNND1 | ARHGAP26 | g.chr11:57,577,695 | g.chr5:142,393,645 |
| ANXA2 | MYO9A | g.chr15:60,656,628 | g.chr15:72,154,952 |
| TKT | RHOA | g.chr3:53,275,148 | g.chr3:49,405,981 |
| ZNF292 | PREX1 | g.chr6:87,923,787 | g.chr20:47,351,184 |
| ECT2 | FABP6 | g.chr3:172,491,812 | g.chr5:159,649,994 |
| EML4 | ALK | g.chr2:42,525,269 | g.chr2:29,446,956 |
| PGAP3 | VMP1 | g.chr17:37,840,850 | g.chr17:57,851,115 |
| TACC2 | PPAPDC1A | g.chr10:123,892,249 | g.chr10:122,263,330 |
| LONP1 | SAFB | g.chr19:5,707,071 | g.chr19:5,645,348 |
| LUC7L3 | C10orf76 | g.chr17:48,819,092 | g.chr10:103,609,649 |
| CLSTN1 | EFCAB7 | g.chr1:9,833,330 | g.chr1:64,027,380 |
| ARFGAP2 | SLC1A2 | g.chr11:47,189,460 | g.chr11:35,287,305 |
| TERF2 | CDH3 | g.chr16:69,395,307 | g.chr16:68,725,623 |
| GTF2I | FBF1 | g.chr7:74,120,764 | g.chr17:73,929,169 |
| ARMC7 | PEX14 | g.chr17:73,106,701 | g.chr1:10,678,389 |
| IFFO2 | UBR4 | g.chr1:19,282,162 | g.chr1:19,412,764 |
| INTS12 | TBCK | g.chr4:106,613,133 | g.chr4:107,114,927 |
| UBE2L3 | MAPK1 | g.chr22:21,965,332 | g.chr22:22,127,271 |
| ELK3 | NTN4 | g.chr12:96,617,551 | g.chr12:96,107,116 |
| EIF4G2 | UPK2 | g.chr11:10,828,367 | g.chr11:118,828,308 |
| RICTOR | GHR | g.chr5:39,074,213 | g.chr5:42,629,140 |
| RNASEH2C | CFL1 | g.chr11:65,487,516 | g.chr11:65,623,713 |
| ARHGAP26 | NDFIP1 | g.chr5:142,311,690 | g.chr5:141,511,373 |
| IDUA | GAK | g.chr4:981,737 | g.chr4:864,692 |

Table 1 In-frame fusions in a discovery set (n = 80)

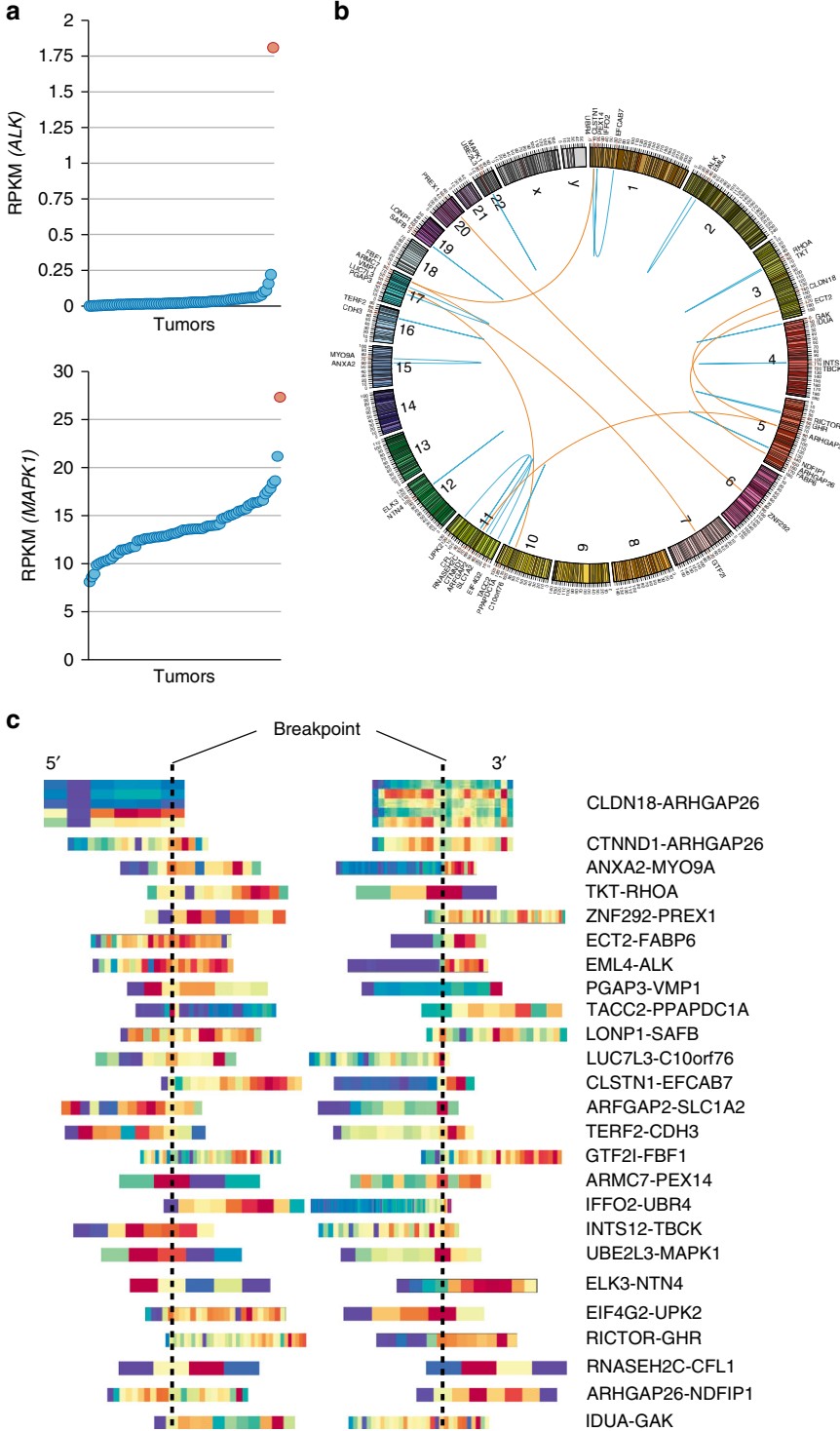

**Fig. 1** Twenty-five in-frame gene fusions identified in a discovery dataset of 80 early-onset DGCs (**a**) Top panel, Expression level of *ALK* in the tumor containing the *EML4-ALK* fusion (magenta circle). Each circle represents each tumor. Sample ordering according to *ALK* expression level. Bottom panel, Expression level of *MAPK1* in the tumor containing the *UBE2L3-MAPK1* fusion (magenta circle). Sample ordering according to *MAPK1* expression level. (**b**) Circos plot for the 25 in-frame fusions. (**c**) Exon-level, Fragments Per Kilobase Of Exon Per Million Fragments Mapped (FPKM) values for the in-frame gene fusions. Heatmap after gene centering. Dotted lines, mRNA breakpoints; Red, higher expression level; Blue, lower expression level. In a majority of fusions, 3′ partner genes overexpressed exons that were located 3′ to the breakpoints

Most of these 25 in-frame fusions overexpressed their 3′ partner genes (Fig. 1c). As with *EML4-ALK*, the *UBE2L3-MAPK1* fusion-containing DGC expressed the highest level of *MAPK1* within the discovery set (Fig. 1a). Similarly, the *ARFGAP2-SLC1A2* fusion-containing DGC expressed the second highest

level of *SLC1A2* (Supplementary Fig. 2). Notably, several studies have reported *SLC1A2*, a glutamate transporter, as a common 3′ partner gene for fusions in solid tumors. For example, *CD44-SLC1A2* fusion is found in both gastric[7] and colorectal cancers[22]. Thus, *SLC1A2* fusions may drive DGC carcinogenesis.

Interestingly, other previously reported clinically-actionable fusions were not present in our discovery set. Fusions involving *BRAF* and *ROS1* have been reported in gastric cancers[6,8], but were not observed in any of the 80 DGCs we analyzed.

DAVID gene ontology analysis revealed that many 3' partner genes of these 25 fusions were significantly enriched within the Rho GTPase pathway (P = 0.01, Supplementary Table 6). Specifically, *ARHGAP26, RHOA, PREX1, MYO9A, ECT2,* and *CFL1* were Rho GTPases or their regulators and downstream effectors. These results suggest that the Rho GTPase pathway is altered by gene fusion events in a subset of DGCs.

**Recurrent in-frame fusions in DGC.** To evaluate the clinical and biological implications of these fusions in a larger sample set of DGCs, we expanded our dataset to include 384 Korean patients. Whereas this expanded dataset included all 80 early-onset DGCs in a discovery dataset, the expanded dataset comprised predominantly (n = 249) of older patients (≥ 46 years of age) with late-onset DGC. We performed RT-PCR on these 384 tumors to analyze the expression of the in-frame fusions previously identified in our early-onset DGCs (Supplementary Table 1). Three of these fusions, *CLDN18-ARHGAP26, CTNND1-ARHGAP26,* and *ANXA2-MYO9A*, were recurrent (*i.e.*, present in ≥ 2 samples) in this expanded dataset (Fig. 2 and Supplementary Fig. 3). Of the 384 DGCs, 17 tumors (4.4%) harbored one of these three in-frame fusions. None of these three fusions were identified in adjacent normal tissue from the corresponding 17 tumors, suggesting that these fusions represent somatic alterations.

The most common fusion, *CLDN18-ARHGAP26*, was significantly more prevalent in early-onset DGC (8 of 135) than late-onset DGC (5 of 249; P = 0.042, chi-square). Of the 13 patients with tumors expressing a *CLDN18-ARHGAP26* fusion, median patient age (38 years) was significantly lower than for the 371 patients without this fusion (54 years; P = 0.042, Wilcoxon; Fig. 2c). Interestingly, *CLDN18-ARHGAP26* was more prevalent among tumors with *H. pylori* than those without (P = 0.034, Fisher's exact test; Supplementary Table 7). Functional studies demonstrate that the ectopic expression of *CLDN18-ARHGAP26* modestly but significantly impaired the aggregation of mouse DGC cell lines (P < 0.001, *t*-test; Fig. 3a and Supplementary Fig. 6), particularly in two cell lines (*Pdx1-cre; Smad4^{F/F}; Trp53^{F/F}; Cdh1^{F/+}* cells and NCC-S1 cells). Given that poorly cohesive cell growth is characteristic of DGC, these findings collectively suggest that relatively high prevalence of the *CLDN18-ARHGAP26* fusion in young DGC patients underlies the strong enrichment of diffuse histology observed in early-onset gastric cancer. By contrast, the ectopic expression of *CLDN18-ARHGAP26* did not enhance the tumorigenic potential of DGC cells (Supplementary Table 8). *ARHGAP26* was also fused to another 5' partner gene, *CTNND1*, that is located at chromosome 11. The mRNA breakpoint position in *ARHGAP26* (g.chr 5:142,393,645) was the same location for both *CTNND1-ARHGAP26* and *CLDN18-ARHGAP26* fusions. Whole genome sequencing (WGS) analysis of a tumor expressing the *CTNND1-ARHGAP26* fusion revealed that g.chr11:57,578,103 (*CTNND1* intron 15) was aberrantly fused to g.chr5:142,358,707 (*ARHGAP26* intron 11; Supplementary Table 9) at the genomic DNA level. The *CTNND1-ARHGAP26* fusion was expressed in 2 of 384 DGCs. Thus, *ARHGAP26* was involved in two distinct interchromosomal translocation events in our expanded DGC dataset at frequencies of 3.4% and 0.1%, for *CLDN18-ARHGAP26* and *CTNND1-ARHGAP26*, respectively.

Another recurrent in-frame fusion, *ANXA2-MYO9A*, was identified in one early-onset and one late-onset case of DGC (Supplementary Figs. 5a and 5c). Our study is the first to report this gene fusion in human cancer tissue samples. In both of the DGCs harboring the *ANXA2-MYO9A* fusion, *ANXA2* exons 1–4 (amino acids 1–99) were fused in-frame to *MYO9A* exons 33–42 (amino acids 1,994–2,548) in the same orientation. Importantly, *MYO9A* exons 33–42 included a RhoGAP domain. The ectopic expression of *ANXA2-MYO9A* in 293FT cells significantly suppressed Rho GTPase relative to the ectopic expression of an empty vector (Fig. 2d and Supplementary Fig. 4). These results suggest the biological relevance of the RhoGAP domain in the pathogenesis of this fusion.

The early-onset DGC containing the *ANXA2-MYO9A* fusion had the highest *MYO9A* expression within the discovery set (Fig. 2e). WGS data of this tumor revealed that g.chr15: 60,656,550 (*ANXA2* intron 4) was aberrantly fused to g.chr15: 72,157,966 (*MYO9A* intron 32; Supplementary Table 9), which was confirmed by PCR sequencing analysis of genomic DNA (Fig. 2f). Such rearrangement was not observed in adjacent normal tissue from the tumor sample, suggesting a somatic alteration (Supplementary Fig. 5b). The tumors expressing the *ANXA2-MYO9A* fusion, an early-onset DGC and a late-onset DGC, demonstrated the stronger cytoplasmic and membranous MYO9A immunostaining than tumors without (P = 0.013, Cochran-Mantel-Haenszel; Fig. 2g and Supplementary Table 10). Thus, the three recurrent fusions each contained a RhoGAP domain in their 3' partner genes. Proteins containing a RhoGAP domain usually function to inactivate RHO family small GTPases[23]. Notably, our mutation analyses revealed that *CDH1* mutations, as well as *RHOA* mutations, were mutually exclusive with expression of the recurrent fusions. The three recurrent in-frame fusions were present in 17 of 384 patients with DGC, yet none of these 17 tumors contained *CDH1* mutations. *CDH1* mutations were present in 31.1% (66 of 212) of sequenced tumors without these fusions (P = 0.006, chi-square). In addition, no *RHOA* mutations were found among these 17 tumors, whereas 15.1% (32 of 212) of sequenced tumors without these fusions (P = 0.08, chi-square) had *RHOA* mutations, demonstrating a trend for mutual exclusivity. As with *CLDN18-ARHGAP26*, the ectopic expression of *Anxa2-Myo9a* impaired the aggregation of mouse DGC cells (*Pdx1-cre; Smad4^{F/F}; Trp53^{F/F}; Cdh1^{F/+}* cells, NCC-S1 cells, and NCC-S1M cells; P < 0.001, *t*-test; Fig. 3b and Supplementary Fig. 6). These data collectively suggest that RhoGAP domain-containing fusions may overlap with and *RHOA*[15] and *CDH1* mutations in functional effect (Supplementary Fig. 7), and may underlie the poorly cohesive growth pattern of a subset of DGCs that are wild-type for *RHOA* and *CDH1*.

**Clinicopathological correlates of RhoGAP domain fusions.** Among the 17 RhoGAP domain fusion-containing DGCs, only one tumor had the MSI and none were EBV-positive. The frequency of fusions did not vary with primary tumor location or gender (Fig. 4a). While gastric cancer TCGA project reported that the *CLDN18-ARHGAP26* fusion is enriched in the GS subgroup[3], we found no difference in the distribution of TCGA molecular subgroup classifications between fusion-positive and fusion-negative DGCs within a larger sample set of DGCs (P = 0.7, chi-square; Fig. 4a).

Patients with RhoGAP domain fusion-containing DGCs (n = 17) had a significantly worse prognosis than those without such fusions (n = 367). The median survival was 29.1 [95% CI, 5.7–not reached] and 94.6 [95% CI, 62.0–not reached] months, respectively (P = 0.011, log-rank; Fig. 4b; HR, 2.8 [95% CI, 1.5–5.3]). Patients with these fusions tended to have more frequent distant metastasis at the time of diagnosis than those without, at 47.2% and 30.5%, respectively. However, this difference was not statistically significant (P = 0.15, chi-square; Fig. 4a). Examining only patients with the *CLDN18-ARHGAP26* fusion (n = 13)

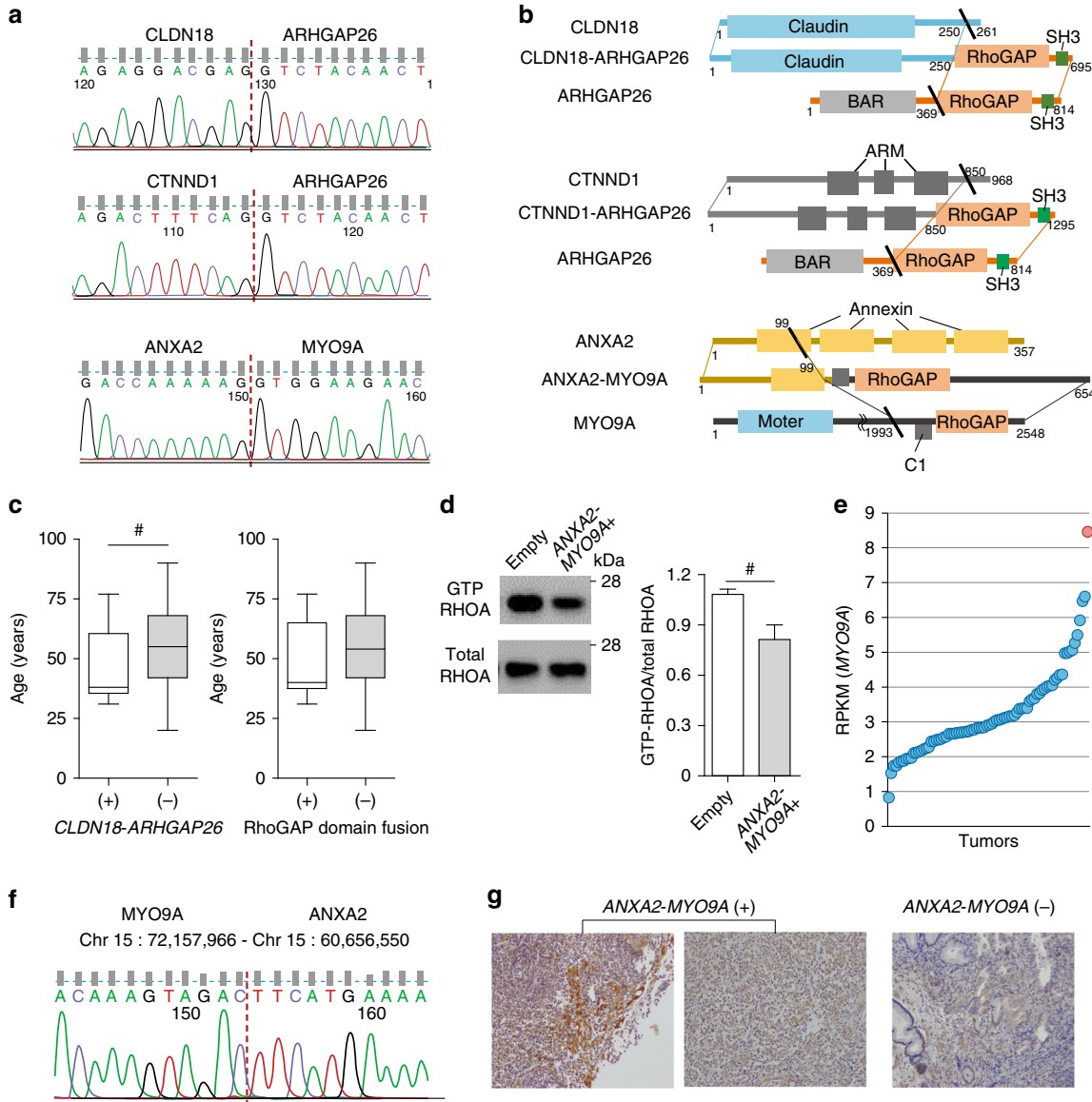

**Fig. 2** Schematic representation of three recurrent RhoGAP domain-containing fusions (**a**) Chromatogram for the RT-PCR sequencing data. Dotted red lines, mRNA breakpoints. (**b**) Top panel, the *CLDN18-ARHGAP26* fusion. Middle panel, the *CTNND1-ARHGAP26* fusion was composed of the Armadillo (Arm) family domain of *CTNND1* that is involved in cell-cell adhesion and signal transduction and RhoGAP and SH3 domains of *ARHGAP26*. Bottom panel, the *ANXA2-MYO9A* fusion was composed of 99 amino acids in the N-terminal portion of *ANXA2* and the protein kinase C conserved region 1 (C1) and RhoGAP domain of *MYO9A*. (**c**) Left panel, Age at the time of diagnosis was significantly lower in patients with the *CLDN18-ARHGAP26* fusion (n = 13; left) than in those without (n = 371; right) (P = 0.042, Wilcoxon). Right panel, Age tended to be lower in patients with RhoGAP domain-containing fusions (n = 17; left) than those without (n = 367; right) (P = 0.10, Wilcoxon). Box plots display 5%, 25%, median, 75%, and 95%. (d) RhoTekin assay in 293FT cells following the ectopic expression of *Anxa2-Myo9a* fusion. Relative value of GTP-RHOA to total RHOA as measured by a densitometry in three independent experiments. Error bars, mean ± SEM. (**e**) RPKM values for *MYO9A* in 80 early-onset DGCs (dots) in a discovery dataset. Each circle represents each tumor. Blue circle, a tumor expressing the *ANXA2-MYO9A* fusion. (**f**) PCR sequencing chromatogram for genomic DNA isolated from a tumor expressing the *ANXA2-MYO9A* fusion. Dotted line, chromosomal breakpoint. (**g**) MYO9A immunohistochemistry images were taken at 200 ×. Left, Two DGCs expressing the *ANXA2-MYO9A* fusion. Right, Representative photographs for a randomly-selected DGC without the *ANXA2-MYO9A* fusion (P = 0.013; *ANXA2-MYO9A*-positive tumors vs. *ANXA2-MYO9A*-negative tumors; Cochran-Mantel-Haenszel). #P < 0.05, *P < 0.01, **P < 0.001, ***P < 0.0001, Error bar, mean ± SEM

revealed a similarly poor prognosis as compared to patients without *CLDN18-ARHGAP26* (Fig. 4c). Specifically, patients with the *CLDN18-ARHGAP26* fusion had a median survival of 26.9 months relative to 94.6 months for those without this fusion (n = 371; P = 0.001, log-rank; Fig. 4c; HR, 2.2 [95% CI, 1.2–4.1]). The prognostic impact of the *CLDN18-ARHGAP26* fusion was comparable to that of the *CDH1* mutation, which is a poor prognostic factor in early-onset DGC[15] (Supplementary Fig. 8

and Supplementary Tables 11 and 12). These results highlight the clinical relevance of RhoGAP-containing fusions, especially the *CLDN18-ARHGAP26* fusion, in DGC.

To gain functional insights into the poor prognostic role of *CLDN18-ARHGAP26* in gastric cancer, we stably expressed *CLDN18-ARHGAP26* in mouse gastric cancer cells. Phosphoproteome profiling analysis revealed that *Regulation of actin cytoskeleton* KEGG pathway phosphoproteins[24–27], such as

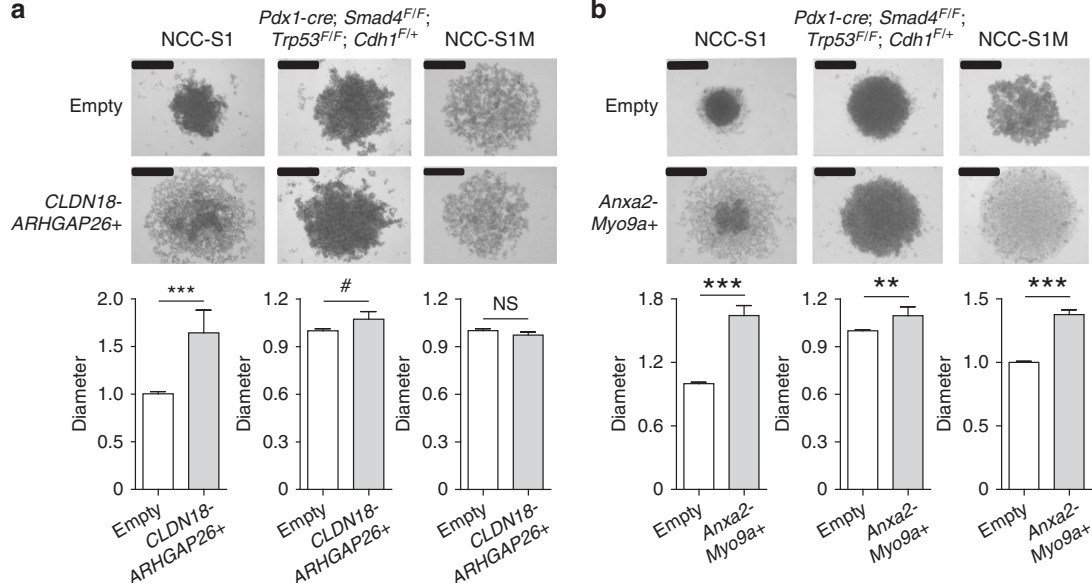

**Fig. 3** Effect of fusions on the aggregation and proliferation of DGC cells (**a**) Slow aggregation assays of mouse DGC cells that stably expressed *CLDN18-ARHGAP26*. Top panels, representative photographs (scale bar = 0.5 mm); Middle panels, cell aggregate diameter of DGC cells that stably expressed *CLDN18-ARHGAP26* relative to cells expressing an empty vector. Average values for at least three independent experiments. Error bars, mean ± SEM. (**b**) Slow aggregation assays of mouse DGC cells that stably expressed *Anxa2-Myo9a*. Top panels, representative photographs (scale bar = 0.5 mm); Middle panels, cell aggregate diameter of DGC cells that stably expressed *Anxa2-Myo9a* relative to cells expressing an empty vector. Average values for at least three independent experiments. Error bars, mean ± SEM. #P < 0.05, *P < 0.01, **P < 0.001, ***P < 0.0001, paired *t*-test

Actin-related protein 2, Actin-related protein 2/3 complex subunit 1B, and Profilin-1, were upregulated by *CLDN18-ARHGAP26* (Supplementary Fig. 9 and Supplementary Table 13). Although mouse gastric cancer cells stably expressing *CLDN18-ARHGAP26* did not significantly differ from empty vector-expressing cells in proliferation, epithelial-mesenchymal transition, tumorigenicity, and sphere-forming capacity (Supplementary Figs.10–14), migration ability was significantly enhanced following *CLDN18-ARHGAP26* overexpression (Fig. 5a and Supplementary Fig. 15a; P < 0.001, Wilcoxon signed rank test). To validate this finding, we evaluated the effect of *CLDN18-ARHGAP26* overexpression on the migration ability of human gastric cancer cells. As in mouse gastric cancer cells, human gastric cancer cells (NUGC4, SNU-719, and SNU-638) stably expressing *CLDN18-ARHGAP26* demonstrated a higher degree of migration ability than those expressing an empty vector (Fig. 5b and Supplementary Figs. 15b and 16; P < 0.001, Wilcoxon signed rank test). Ectopic expression of *CLDN18-ARHGAP26* also enhanced the invasion capacity of these human gastric cancer cells (Fig. 5c; P < 0.001, Wilcoxon signed rank test). These functional data suggest that *CLDN18-ARHGAP26* confers the metastatic phenotype on gastric cancer by enhancing migration and invasion capacities.

**TACC2-PPAPDC1A identified as another recurrent fusion**. To extend our initial findings from RNA sequencing and RT-PCR analyses, we then conducted targeted RNA sequencing analyses for all exons from the 25 in-frame gene fusions first identified by RNA sequencing analysis of the discovery set. The expanded dataset DGCs without available RNA sequencing data (n = 225) were subjected to targeted RNA sequencing analyses, with the mean sequencing coverage of 56.1-fold. We identified additional novel gene fusions harboring mRNA breakpoints that were different from those initially discovered (Table 2). In the 225 DGCs, targeted RNA sequencing analyses revealed three additional

*CLDN18-ARHGAP26* fusion events, two of which had mRNA breakpoints distinct from breakpoints identified by RNA sequencing analysis of the discovery set. More importantly, targeted RNA sequencing identified two additional *TACC2-PPAPDC1A* fusion events with an mRNA breakpoint different from the breakpoint initially discovered (Fig. 6a). Thus, we identified a novel recurrent in-frame fusion present in 1% of tumors (3 of 305) sequenced by either RNA sequencing or targeted RNA sequencing analyses. Our study is the first to report *TACC2-PPAPDC1A* as a recurrent fusion gene in human cancer tissue samples, although *PVT1-PPAPDC1A* has been reported to be present in a gastric cancer cell line[20]. In the early-onset, discovery set DGC harboring *TACC2-PPAPDC1A*, *TACC2* exons 1–6 (amino acids 1–1899) were fused in-frame to *PPAPDC1A* exons 2–7 (amino acids 20–271) in the same orientation. In the other two DGCs with this fusion, *TACC2* exons 1–3 (amino acids 1–48) were also fused in-frame to *PPAPDC1A* exons 2–7 (amino acids 20–271) in the same orientation. No *TACC2-PPAPDC1A* transcripts were present in normal adjacent tissue of the three tumors, suggesting their somatic nature.

RNA sequencing analysis revealed that the early-onset DGC containing *TACC2-PPAPDC1A* displayed the highest *PPAPDC1A* expression and expressed higher levels of *PPAPDC1A* mRNA than tumors with 10q26.1 gene amplification (Fig. 6b). Tissue distribution of *PPAPDC1A* (*DPPL2*) mRNA expression is restricted mainly to the brain, kidney and testes, with no endogenous *PPAPDC1A* expression in the stomach[28]. In DGCs carrying *TACC2-PPAPDC1A*, *PPAPDC1A* overexpression was presumably driven by the *TACC2* promoter, as a result of the in-frame fusion event. Mass spectrometry-based, lipidomic profiling analyses[29] suggested relatively high phospholipid phosphatase activity in the early-onset DGC expressing *TACC2-PPAPDC1A* (P = 0.01, one-sample rank sum test; Fig. 6c and Supplementary Fig. 17). This finding suggests that the possible functional relevance of the *TACC2-PPAPDC1A*, although further biochemical validation is needed.

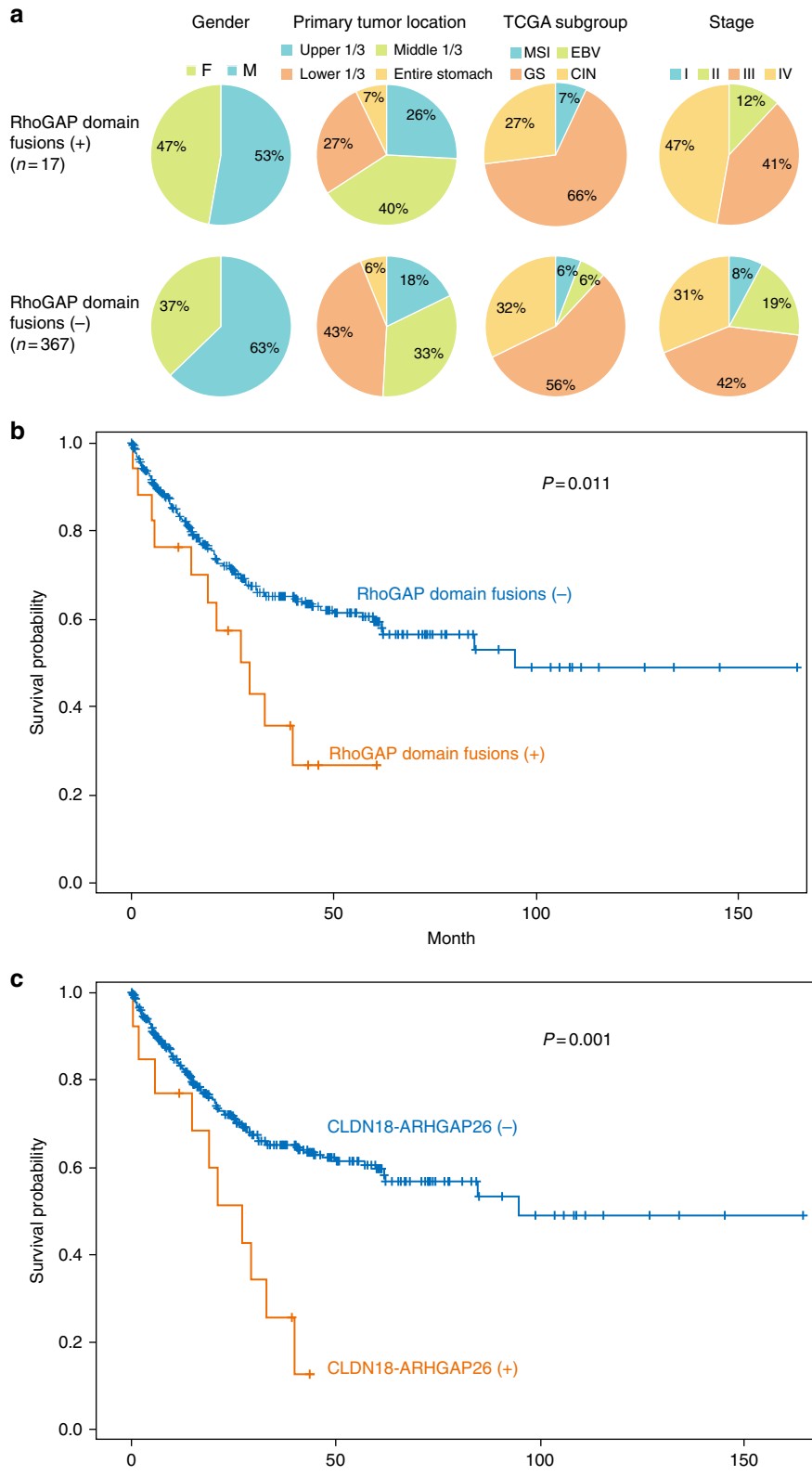

**Fig. 4** Clinical implications of RhoGAP domain-containing fusions in the expanded dataset (n = 384) (**a**) Clinicopathological characteristics (gender and primary tumor location) and TCGA subgroups in patients with (upper row) and without (lower row) RhoGAP domain-containing fusions. (**b**) Kaplan-Meier curves for the overall survival of patients with DGCs expressing three recurrent RhoGAP domain-containing fusions (n = 17) and those without (n = 367). (**c**) Kaplan-Meier curves for the overall survival of patients with DGCs expressing the *CLDN18-ARHGAP26* fusion (n = 13) and those without (n = 371)

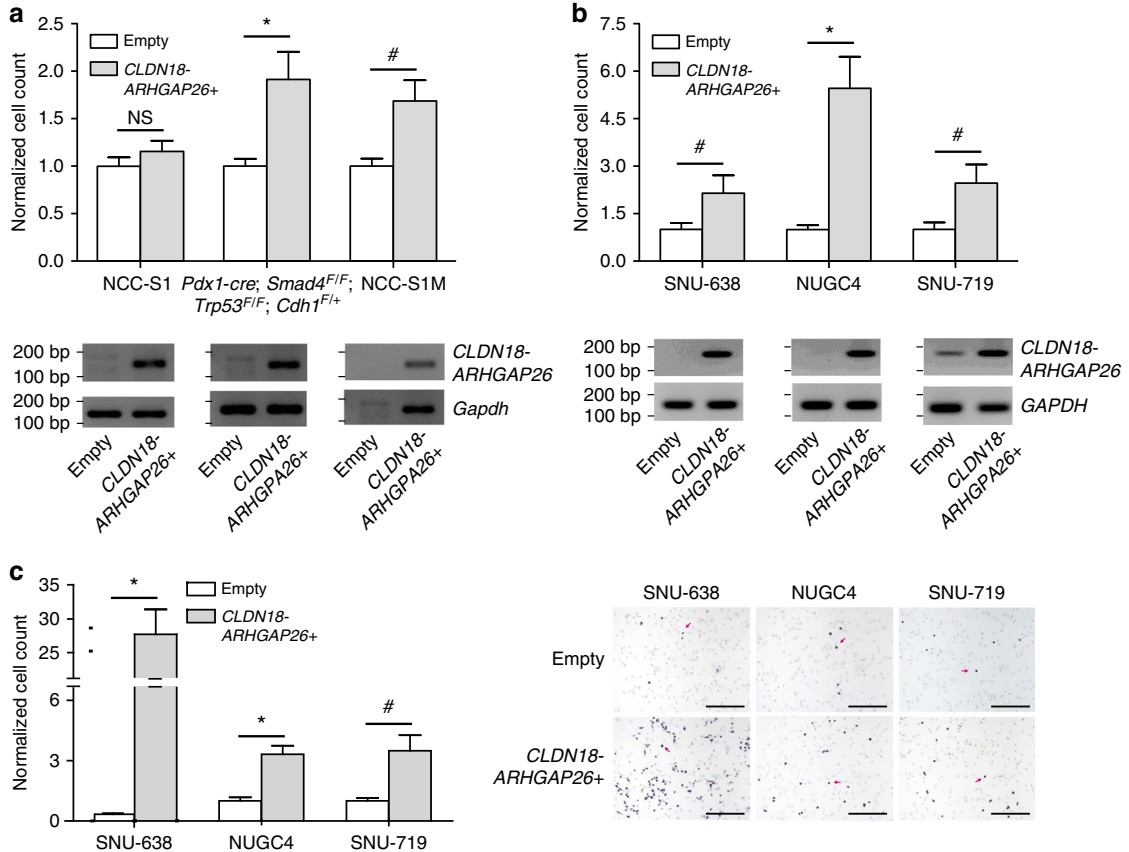

**Fig. 5** Functional effects of *CLDN18-ARHGAP26* overexpression (**a**) Top, Trans-well migration assay. Mouse gastric cancer cells stably expressing *CLDN18-ARHGAP26* demonstrated significantly enhanced migration ability compared with those expressing an empty vector (P < 0.001, Wilcoxon signed rank), and this effect was prominent in NCC-S1M and *Pdx1-cre;Smad4$^{F/F}$;Trp53$^{F/F}$;Cdh1$^{F/+}$* cells. Vertical axis denotes the normalized number of migrated cells per field. Average values of 4 independent experiments; Bottom, RT-PCR for *CLDN18-ARHGAP26* (**b**) Top, Migration assay of human gastric cancer cells following the ectopic expression of *CLDN18-ARHGAP26*. Human gastric cancer cells stably *CLDN18-ARHGAP26* demonstrated significantly enhanced migration capacity compared with those expressing an empty vector (P < 0.001, Wilcoxon signed rank). Vertical axis denotes the normalized number of migrated cells per field. Average values of 3 independent experiments; Bottom, RT-PCR for *CLDN18-ARHGAP26*. (**c**) Left, Invasion assay. Human gastric cancer cells stably expressing *CLDN18-ARHGAP26* demonstrated significantly enhanced invasion capacity compared with those expressing an empty vector (P < 0.001, Wilcoxon signed rank). Vertical axis denotes the normalized number of migrated cells per field. Average values of three independent experiments. Right, Representative photographs of the invasion assay (200 ×, scale bar = 0.2 mm). Red arrows, gastric cancer cells. Error bars, mean ± SEM. $^{#}$P < 0.05, *P < 0.01, **P < 0.001, ***P < 0.0001

All three patients with *TACC2-PPAPDC1A* gene fusions were male, and their median age (67 years) did not differ from the 302 sequenced dataset DGCs without the fusion (P = 0.51, *t*-test). One of the three tumors belonged to the CIN subgroup, while the other two tumors were in the GS subgroup. Thus, the distribution of TCGA subgroup and primary tumor location was not different according to *TACC2-PPAPDC1A* status. Patients with *TACC2-PPAPDC1A* tended to exhibit more frequent distant metastasis at the time of diagnosis (66.6%) than those without (24.5%; P = 0.15, Fisher's exact test; Fig. 7a). Consistently, patients with DGCs expressing *TACC2-PPAPDC1A* had a significantly worse prognosis than those without such fusions. The median survival time for those with *TACC2-PPAPDC1A* was 3.5 months [95% CI, 0.5–not reached], while a median survival time was not reached among those without [95% CI, 61.8–not reached; P < 0.0001, log-rank; HR, 8.6 [95% CI, 2.7–27.3]; Fig. 7b). Two of the three tumors harbored gene amplifications in the 10q26.1 locus, whereas the third tumor contained *CDH1* mutations. We observed, however, that the prognostic impact of *TACC2-PPAPDC1A* was independent of 10q26.1 amplification and *CDH1* mutations (adjusted HR, 7.1 [95% CI, 2.1–23.6]; Supplementary Table 14).

In addition to *TACC2*, another in-frame fusion event with *PPAPDC1A* was observed with *ARIH1* as the 5' partner gene. In this DGC, *ARIH1* exons 1–2 (amino acids 1–147) at chromosome 15 were fused in-frame to *PPAPDC1A* exons 2–7 (amino acids 20–271) at chromosome 10 (Fig. 6a). Thus, all four *PPAPDC1A* fusion events included the PAP2 (type 2 phosphatidic acid phosphatase) domain as the 3' partner gene, and no *PPAPDC1A* fusion transcripts were present in normal adjacent tissue. When all four *PPAPDC1A* fusions were considered, *PPAPDC1A* continued to be a strong prognostic indicator, independent from 10q26.1 gene amplification (adjusted HR, 7.8 [95% CI, 2.3–25.8]; Fig. 7c).

**Actionable FGFR2-TACC2 identified by targeted RNA sequencing.** Interestingly, 5 of 18 in-frame fusions identified by our targeted RNA sequencing were located at the chromosome locus 10q26.1 (Table 2; P = 0.002, chi-square). One such novel in-frame fusion involving this locus was *FGFR2-TACC2*. Given that *FGFR1-TACC1* and *FGFR3-TACC3* play oncogenic roles in several solid tumors[30], this novel fusion might contribute to DGC carcinogenesis through activation of the FGFR2 kinase mediated

**Table 2 In-frame gene fusions additionally identified by targeted RNA sequencing analysis**

| Gene name | | mRNA Breakpoint | | Domain included |
|---|---|---|---|---|
| 5' gene | 3' gene | 5' gene | 3' gene | |
| TACC2 | PPAPDC1A | g.chr10:123,810,065 | g.chr10:122,263,330 | PAP2[a] (3') |
| TACC2 | PPAPDC1A | g.chr10:123,810,065 | g.chr10:122,263,330 | PAP2 (3') |
| ARIH1 | PPAPDC1A | g.chr15:72,810,475 | g.chr10:122,263,330 | PAP2 (3') |
| FGFR2 | TACC2 | g.chr10:123,243,212 | g.chr10:123,996,910 | |
| SEC23IP | TACC2 | g.chr10:121,680,496 | g.chr10:123,996,910 | |
| TACC2 | WDR11 | g.chr10:123,976,343 | g.chr10:122,630,682 | |
| VMP1 | RPS6KB1 | g.chr17:57,915,758 | g.chr17:57,987,923 | |
| PGAP3 | PSMD3 | g.chr17:37,840,850 | g.chr17:38,144,936 | |
| PGAP3 | PIP4K2B | g.chr17:37,840,850 | g.chr17:36,927,525 | |
| NFAT5 | TERF2 | g.chr16:69,693,802 | g.chr16:69,419,389 | |
| ECT2 | FNDC3B | g.chr3:172,520,770 | g.chr3:171,965,323 | |
| CLDN18 | ARHGAP6 | g.chr3:137,749,947 | g.chrX:11,272,827 | RhoGAP (3') |
| ARHGAP26 | CAST | g.chr5:142,311,690 | g.chr5:96,089,764 | |
| ARHGAP26 | GLRA1 | g.chr5:142,311,690 | g.chr5:151,239,569 | |
| WDR7 | ARHGAP26 | g.chr18:54,448,887 | g.chr5:142,252,965 | RhoGAP (3') |
| CLDN18 | ARHGAP26 | g.chr3:137,749,947 | g.chr5:142,393,645 | RhoGAP (3') |
| CLDN18 | ARHGAP26 | g.chr3:137,749,946 | g.chr5:142,292,764 | RhoGAP (3') |
| CLDN18 | ARHGAP26 | g.chr3:137,749,948 | g.chr5:142,292,765 | RhoGAP (3') |

[a]PAP2, type 2 phosphatidic acid phosphatase

by the coiled-coil domain of TACC2. Notably, *FGFR2-TACC2* is clinically-actionable, similar to *EML4-ALK* as described above. Thus, our comprehensive in-frame fusion screen determined that 0.7% (2 of 305) of the sequenced dataset DGCs harbor clinically-actionable fusions. Table 2 summarizes in-frame gene fusions additionally identified by targeted RNA sequencing analysis. All fusions, except *CLDN18-ARHGAP26*, represent novel discoveries related to DGCs. Included in this list was *VMP1-RPS6KB1*, which was previously reported in breast cancer and esophageal adenocarcinomas[21,31], but not in gastric cancer.

**Recurrent fusions defined the subgroup with worst prognosis**. Our comprehensive in-frame fusion screen using the largest-ever set of DGCs identified the *TACC2-PPAPDC1A* and RhoGAP domain-containing in-frame fusions as recurrent somatic alterations. Presence of one of these recurrent in-frame fusions (*TACC2-PPAPDC1A*, *CLDN18-ARHGAP26*, *CTNND1-ARHGAP26*, or *ANXA2-MYO9A*) was associated with very poor prognosis. The median survival time for those with recurrent fusions was 22.8 months [95% CI, 5.0–33.0], while a median survival time was not reached among those without recurrent fusions (P < 0.0001, log-rank; HR, 3.5 [95% CI, 2.0–6.0]). Univariate survival analyses indicated that chromosomal instability (CIN), which was determined by parallel SNP6.0 array analyses[15], and *CDH1* mutation were significant prognostic genomic alterations in our DGCs, consistent with our previous report[15] (HRs, 2.3 [95% CI, 1.3–3.9] and 1.8 [95% CI, 1.1–3.0], respectively), whereas *TP53* mutations were not (P = 0.80, log rank). According to Cox regression analysis, the prognostic impact of recurrent in-frame fusions (adjusted HR, 4.3 [95% CI, 2.1–8.4]) was independent of the chromosomal instability and the *CDH1* mutation (Table 3). Overall, the *PPAPDC1A* and RhoGAP domain-containing in-frame fusions were present in 7.5% (23 of 305) of the sequenced dataset DGCs, but not in adjacent normal tissue. These fusions clearly defined the subset of aggressive DGCs (HR, 3.4 [95% CI, 2.0–5.7]; Fig. 7d), and their prognostic impact was higher than, and independent of, the chromosomal instability and the *CDH1* mutation (Table 3).

## Discussion

In this study, we investigated the biological and clinical implications of fusion genes in early-onset DGCs. According to RNA sequencing and RT-PCR analyses, three in-frame fusions were recurrent. All these three contained a RhoGAP domain[23] in their 3' region (Fig. 2b), suggesting that this domain may have biological relevance to the pathogenesis of fusions in DGC. The RHO family is comprised of small G proteins that are inactivated by GTPase-activating proteins by stimulating the intrinsic GTPase activity of small G proteins. The C-terminal end of *ARHGAP26* and the effector region of *MYO9A* were conserved in fusion genes, suggesting that the Rho family GTPase pathway is a primary target of recurrent in-frame fusion transcripts in DGC[3]. This study is the first to demonstrate that the three RhoGAP domain-containing fusions were mutually exclusive with *CDH1* mutations in DGC. Previous reports have suggested that the *CLDN18-ARHGAP26* fusion is mutually exclusive with *RHOA* mutations[3], which corresponds with observations in the current study. Mutations in *CDH1* and *RHOA* impair cell adhesion in a process characteristic of DGC pathogenesis. Given that these fusions were mutually exclusive with *CDH1* and *RHOA* mutations but impaired cell aggregations in a manner similar to such mutations, RhoGAP domain-containing fusions may play a role in the development of the poorly cohesive growth pattern characteristic of DGC.

Interestingly, the *CLDN18-ARHGAP26* fusion was significantly more common in younger patients than in older patients, as with the *ALK* or *RET* fusions in lung adenocarcinomas[10,11]. A majority of gastric cancers in young patients exhibits diffuse type histology, and RhoGAP domain-containing recurrent fusions were more prevalent in the diffuse type than in the intestinal type (P = 0.03, chi-square; Supplementary Table 15). We also observed a trend for the relatively high prevalence of the *CLDN18-ARHGAP26* fusion among *H. pylori*-positive DGCs. The association between *H. pylori* and chromosomal translocation has not been reported in gastric adenocarcinomas, unlike in MALT lymphomas. Further studies are warranted to validate this interesting finding.

Another novel recurrent fusion, *ANXA2-MYO9A*, resulted from an intrachromosomal rearrangement and led to the

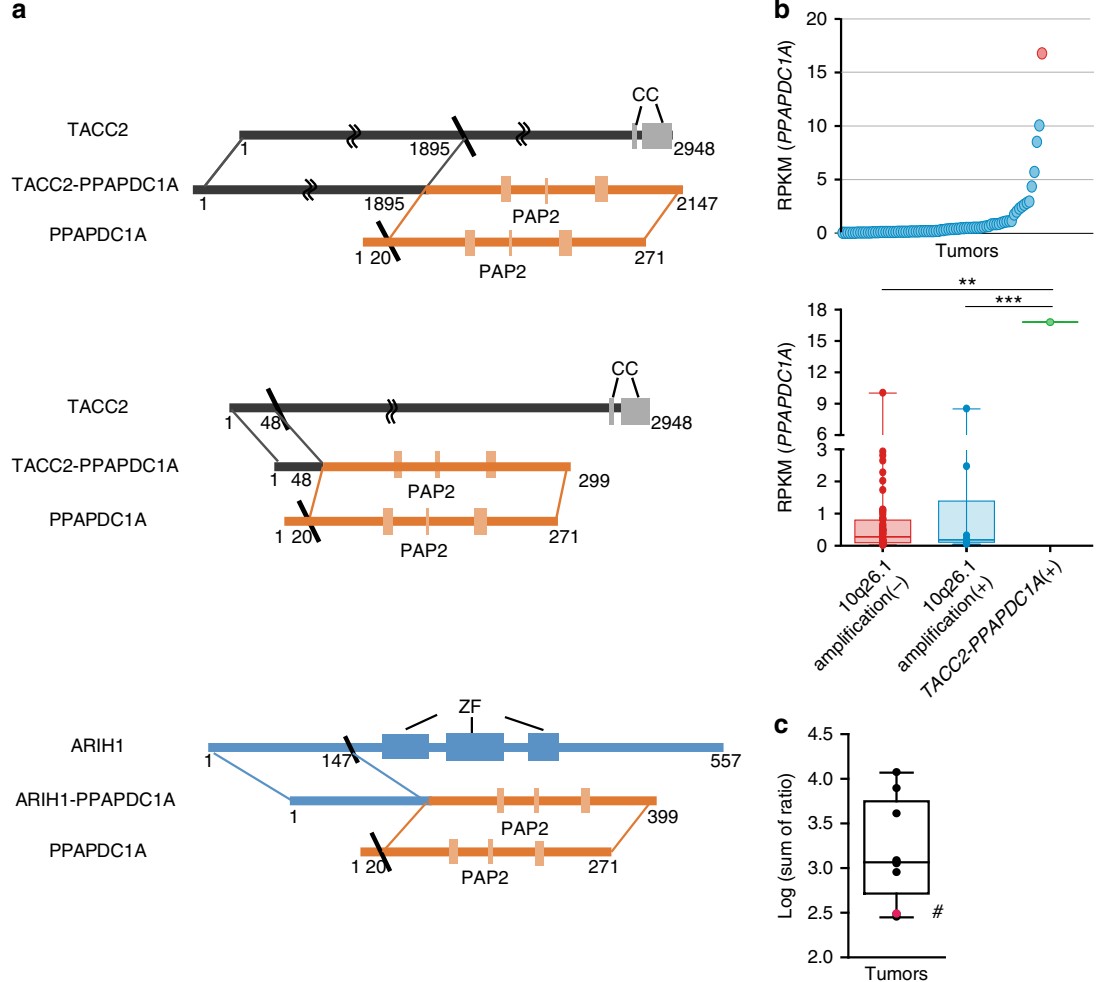

**Fig. 6** The *PPAPDC1A* in-frame fusions identified by targeted RNA sequencing (**a**) Schematic representation of the *PPAPDC1A* fusions. Top and middle panels, *TACC2-PPAPDC1A* fusions; Bottom panel, *ARIH1-PPAPDC1A*. CC, Coiled-coil domain; PAP2, PAP2 domain; ZF, Zinc finger domain. (**b**) Left, *PPAPDC1A* RPKM values for the 80 discovery set DGCs. Each circle represents each tumor. Sample ordering according to the *PPAPDC1A* RPKM. Magenta circle, early-onset DGC expressing *TACC2- PPAPDC1A*. Right, *PPAPDC1A* RPKM values for the discovery set DGCs with and without 10q26.1 gene amplification. Turquoise circle, early-onset DGC expressing *TACC2- PPAPDC1A*. Box plots display 5%, 25%, median, 75%, and 95%. (**c**) Lipidomic profiling data for DGCs with or without *TACC2-PPAPDC1A*. The sum of the mass spectral peak area ratio between phosphatidic acid (PA, the substrate of PPAPDC1A) and diacylglycerol (DG, the product of PPAPDC1A) was significantly lower in an early-onset DGC expressing *TACC2-PPAPDC1A* in the discovery set than in randomly-selected eight DGCs without the fusion (P = 0.01, one-sample rank sum test). Vertical axis denotes the sum of mass spectral peak area ratio between PA and the corresponding DG for each PA-DG pair. Each circle represents each tumor. Magenta circle, early-onset DGC expressing *TACC2- PPAPDC1A*. #P value for an one-sample rank sum test. Box plots display 5%, 25%, median, 75%, and 95%

overexpression of its 3' partner gene, *MYO9A*. Myosin 9 A (Myr7) contains the RhoGAP domain and is an actin-dependent motor protein of the unconventional myosin IX class. The Rho-GAP domain enables class IX myosins to inactivate small GTPases of the Rho family[32]. Specifically, Myr7 inactivates Rho by stimulating its GTPase activity in neurons[23,33]. *MYO9A* is expressed in several tissues and is enriched in the brain and testes[33,34]. *Myo9a* knockout mice develop hydrocephalus and kidney dysfunction, which highlights the importance of *MYO9A* in epithelial cell morphology and differentiation[35,36]. *ANXA2* encodes Annexin A2, a calcium-regulated membrane-binding protein[37]. *MYO9A* was overexpressed in the tumor containing the *ANXA2-MYO9A* fusion, which supports transcriptional activation as the oncogenic mechanism for this gene fusion.

Our cell aggregation data regarding ectopic expression of *CLDN18-ARHGAP26* are consistent with Yao et al.'s data studying the effect of this same fusion in HGC27 cells[9]. Yao et al. suggested that *CLDN18-ARHGAP26* fusion compromises the role of *CLDN18*

in epithelial barrier promotion and directly affects the intactness of the paracellular barrier[9]. Given our RNA sequencing and RT-PCR data that all recurrent in-frame fusions contain RhoGAP domain, functional alteration of *ARHGAP26* might possibly be more important than that of impaired *CLDN18* function in the oncogenic mechanism of *CLDN18-ARHGAP26* fusion. Given that cell migration/invasion activities are regulated by complex crosstalk between RHO GTPases[38–40], further biochemical studies are warranted to explore how *CLDN18-ARHGAP26* affect the migration/invasion activities of gastric cancer cells.

Our targeted RNA sequencing analysis[41] was the first to reveal that DGC's in-frame fusion events frequently involved the chromosomal fragile site at 10q26.1 that harbors *TACC2*, *FGFR2*, and *PPAPDC1A*, in addition to RhoGAP domain-containing genes. This result may be consistent with our previously reported genomic data, which identified the chromosomal locus 10q26.1 as the most recurrent focal gene amplification in Korean DGCs[15]. Although the 10q26.1-amplified gastric cancer cell line SNU16

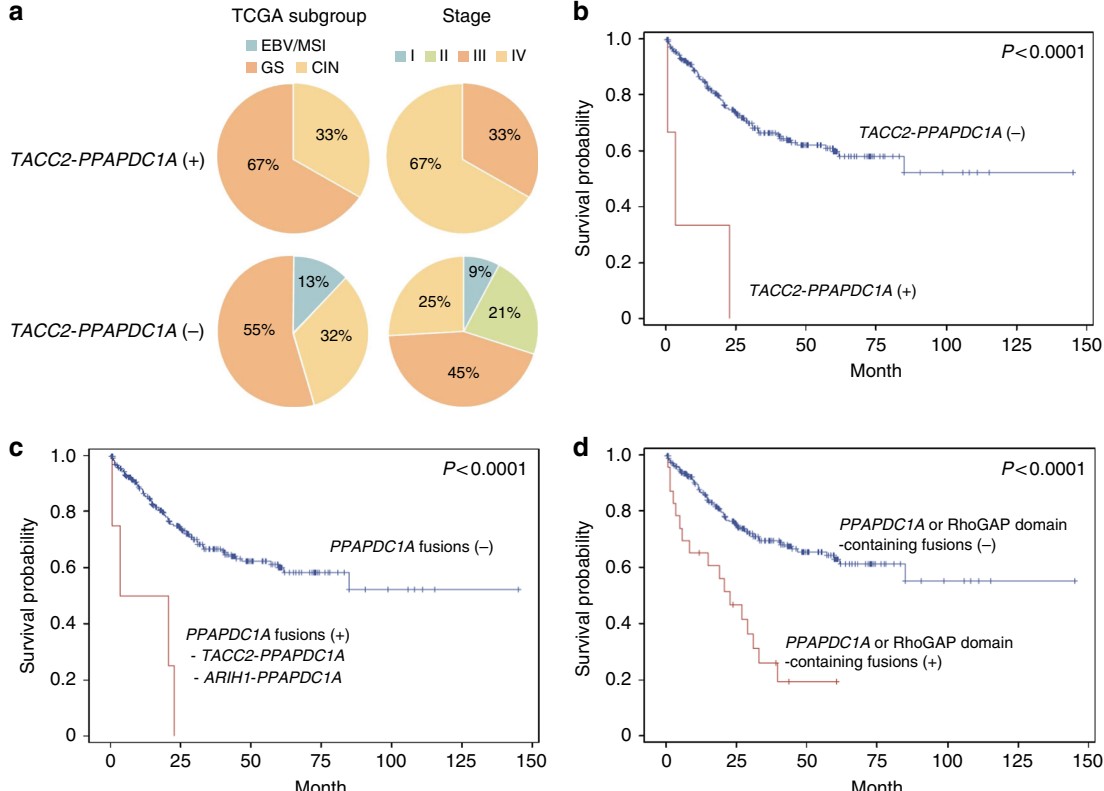

**Fig. 7** Clinical implications of the *TACC2-PPAPDC1A* fusion in the sequenced dataset (n = 305) (**a**) Stage at the time of diagnosis and TCGA subgroups in patients with or without *TACC2-PPAPDC1A*. TCGA subgroup was determined only in DGCs with available SNP6.0 data. (**b**) Kaplan-Meier curves for the overall survival of three patients with DGCs expressing *TACC2-PPAPDC1A* and those without (n = 302). Their median survival times were 3.5 months and not reached, respectively (P < 0.0001, log-rank). (**c**) Kaplan-Meier curves for the overall survival of four patients with DGCs expressing *PPAPDC1A* fusions and those without (n = 301). Their median survival times were 12.1 months and not reached, respectively (P < 0.0001, log-rank; HR, 6.6 [95% CI, 2.4–18.1]). (**d**) Kaplan-Meier curves for the overall survival of 23 patients with the *PPAPDC1A* or RhoGAP domain-containing fusions and those without (n = 282). Their median survival times were 22.8 months [95% CI, 5.7–33.0] and not reached, respectively (P < 0.0001, log-rank; HR, 3.4 [95% CI, 2.0–5.7])

| Table 3 Cox regression survival analysis of 173 DGCs with available SNP6.0 array and DNA sequencing data | | | |
|---|---|---|---|
| **Analysis of recurrent fusions** | ***n*** | **P** | **HR[a] (95% CI)** |
| Recurrent fusions[b] | 18 | < 0.0001 | 4.3 (2.1–8.4) |
| CIN[c] | 55 | 0.002 | 2.4 (1.4–4.1) |
| *CDH1* mutation | 57 | 0.004 | 2.5 (1.3–4.5) |
| **Analysis of *PPAPDC1A* or RhoGAP domain fusions** | ***n*** | **P** | **HR[a] (95% CI)** |
| *PPAPDC1A* or RhoGAP domain-containing fusions[d] | 20 | < 0.0001 | 4.3 (2.2–8.3) |
| CIN[c] | 55 | 0.001 | 2.5 (1.4–4.3) |
| *CDH1* mutation | 57 | 0.003 | 2.5 (1.4–4.7) |

[a]HR: hazard ratio
[b]Recurrent fusions: presence of *TACC2-PPAPDC1A*, *CLDN18-ARHGAP26*, *CTNND1-ARHGAP26*, or *ANXA2-MYO9A*
[c]CIN: chromosomal instability
[d]PPAPDC1A or RhoGAP domain fusions: presence of *TACC2-PPAPDC1A*, *ARIH1-PPAPDC1A*, *CLDN18-ARHGAP26*, *CTNND1-ARHGAP26*, *ANXA2-MYO9A*, *WDR7-ARHGAP26*, or *CLDN18-ARHGAP6*

expresses *PVT1-PPAPDC1A*[20], *PPAPDC1A* fusion events have not been reported recurrent in human gastric cancers. While our analysis was limited by the relatively small number of *PPAPDC1A* fusion events in our dataset, the prognostic impact of *PPAPDC1A* fusions was higher than, and independent of, that of 10q26.1 gene amplification. *PPAPDC1A* encodes a phospholipid phosphatase that converts phosphatidic acids to diacylglycerols[42], and our lipidomic profiling data suggested the biochemical relevance of *TACC2-PPAPDC1A* mRNA expression. While we cannot rule out a possibility that *PPAPDC1A* fusions may just represent the

genomic instability of DGCs, our findings warrant further functional studies that evaluate potentiality of *PPAPDC1A* fusions' use as therapeutic targets such as neoantigens in personalized immunotherapy.

Compared to intestinal-type gastric cancer, DGC has not been fully investigated for prognostic factors despite its significant worldwide disease burden[1,2]. We previously reported that CIN, followed by *CDH1* alteration, is the most adverse prognostic factor in early-onset DGCs[15]. Our current, comprehensive study systematically explored recurrent in-frame somatic fusions,

excluding less clinically relevant fusion events such as read-through transcripts. In addition, rigorous RT-PCR analyses of tumors and adjacent normal tissue validated the somatic nature of novel in-frame fusions. Similar genomic studies have not yet been conducted on adequately-sized DGC sets with long-term follow up. Thus, here we present the first genomic study that clearly demonstrates the prognostic impact of novel recurrent *PPAPDC1A* and RhoGAP domain-containing fusions, which was more prominent than those of the chromosomal instability and *CDH1* mutations. These fusions partially overlapped with CIN tumors, but their prognostic impact was independent of that of CIN. In summary, our findings suggest possible roles that Rho-GAP and PAP2 domains play in cancer progression and provide novel genomic insights guiding future strategies for managing DGCs.

## Methods

**Patients**. This study was approved by the National Cancer Center Institutional Review Board (IRB; NCCNCS-120581), and all patients signed IRB-approved consent forms. RNA sequencing analyses were performed in 80 resected tumors and 65 normal tissue adjacent to the 80 tumors that had RNA sequencing data. These tissue samples were collected from patients with early-onset ( ≤ 45 years) DGCs (discovery dataset). For RT-PCR analysis of gene fusions, the dataset was expanded to 384 biopsy and surgical DGC samples (Supplementary Table 1). To estimate the sample size required for an expanded dataset, we hypothesized that recurrent in-frame fusions are present in 15% of tumors and adversely affect prognosis by a hazard ratio of 2. At two-tailed α and β errors of 0.05 and 0.2, respectively, 128 events were estimated to be required to evaluate the effect of fusions on survival[43]. We assumed that about one third of patients with advanced stage gastric cancers die during 3-year follow-up[44]. For 128 events, therefore, 384 tumors were estimated to be required as an expanded dataset.

**RNA sequencing and the identification of novel fusions**. Transcriptome libraries were prepared using poly(A) + RNA isolated from 1–2 μg of total RNA from frozen macrodissected tumors in a discovery dataset and TruSeq mRNA Kit (Illumina, San Diego, CA). Paired-end libraries were sequenced using an Illumina HiSeq 2000 instrument (2 × 100 nucleotide read length). RNA-seq Data Analysis (PRADA)[45,46] was used for fusion discovery. Using the preprocess module of PRADA, we aligned RNA sequencing reads on human reference genome hg19/GRCh37 and human transcripts of Ensembl build 64. We discovered gene fusion candidates using the fusion module, and selected in-frame fusion candidates using the prada-frame utility. In parallel, deFuse[47], FusionMap[48], and TopHat-Fusion[49] were additionally used to predict candidate fusion breakpoints (Supplementary Fig. 19). To remove false-positive breakpoints resulting from these algorithms, we reconstructed candidate regions containing putative breakpoints using Trans-ABySS[50,51] (v1.4.4). Trans-ABySS performed a de novo assembly on candidate regions containing putative breakpoints that were identified by deFuse, FusionMap and TopHat-Fusion. We then performed RT-PCR sequencing analyses to validate the expression of candidate fusions that were in-frame and contained partner genes of importance based on the data in the literature (Supplementary Fig. 19 and Supplementary Table 5).

**RT-PCR analysis of mRNA breakpoints and mutation analyses**. Total RNA samples from the expanded dataset were subjected to RT-PCR analyses of 25 validated in-frame fusions. Synthesized cDNA was PCR-amplified for 35 cycles. Mutations in *CDH1, RHOA* and *TP53* were identified by targeted DNA sequencing analyses[15].

**Cell lines and lentiviral vectors**. NCC-S1 and NCC-S1M cell lines were established by our group from a diffuse-type gastric adenocarcinoma formed in a *Villin-cre; Smad4^{F/F}; Trp53^{F/F}; Cdh1^{F/+}* mouse[52,53]. *Pdx1-cre;Smad4^{F/F};Trp53^{F/F};Cdh1^{F/+}* cells were primary cultured by our group from a diffuse-type gastric adenocarcinoma formed in a *Pdx1-cre; Smad4^{F/F}; Trp53^{F/F}; Cdh1^{F/+}* mouse[52,53]. To generate the *Anxa2-Myo9a* lentiviral expression vector, mouse genomic DNA sequences that were homologous to human *ANXA2-MYO9A* (*Anxa2* coding sequence 1–300 and *Myo9a* coding sequence 6,694–7,896) were synthesized and subcloned using a pCDH-CAG-MSC-EF1-Neomycin vector. For the *CLDN18-ARHGAP26* lentiviral vector, *CLDN18* coding sequence 1–750 and *ARHGAP26* coding sequence 1,108–2,280 were ligated and cloned using a pCDH-CAG-MSC-EF1-Puromycin vector.

**Targeted RNA sequencing analysis**. Of the 304 expanded dataset DGCs without available RNA sequencing data, 225 tumors (74%) were subjected to a hybrid capture-based, custom targeted RNA sequencing analysis[38]. These 225 DGCs were combined with the 80 DGCs with available RNA sequencing data and this combined dataset (sequenced dataset (n = 305)) was used to determine frequencies of *PPAPDC1A* fusions (Supplementary Table 2). Hybrid capture probes were designed to cover all the exons of 25 in-frame fusions listed in Table 1. SureSelect RNA Direct (Agilent Technologies, Santa Clara, CA) was used for library construction. Gene fusions were identified using deFuse[47] with default parameters. In-frame fusion transcripts with > 3 spanning reads were identified as fusion candidates, and were confirmed by RT-PCR sequencing analyses (Table 2).

**Code availability**. The source code of a program to predict if an RNA sequence is in frame is available from the corresponding author on reasonable request.

## Data availability
Our genomic data are deposited to the European Genome-phenome Archive database (EGAD00001002187, EGAD00010000889, and EGAD00001003953) and to the Gene Expression Omnibus (GSE110875).

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

## Acknowledgements

This work was supported by the National Cancer Center Grants 1810950/1710890/1510110, and by the Multi-omic Research Program funded by the Ministry of Science, ICT, and Future Planning in Korea. The biospecimens for this study were provided by members of National Biobank of Korea (Ajou Human Bio-Resource Bank, Asan Bio-Resource Center, Keimyung Human Bio-Resource Bank, Biobank of Pusan National University, Biobank of Chonnam National University Hwasun Hospital, and Biobank of Chungnam National University Hospital), which is supported by the Ministry of Health and Welfare, by Resource Banks at Dong-A University Medical Center and Kosin University Gospel Hospital and by the National Cancer Center of Korea. We thank all the patients and their families who contributed to this study.

## Author contributions

H.Y., W.R.K., J.H.K., K.S.C. and J.W.K. conducted DNA/RNA/protein and functional analyses and wrote a manuscript. D.W.H., S.Y.C., J.L., N.T., A.H. and A.M. performed bioinformatics analyses and wrote a manuscript. Y.S.P., H.H., S.K., S.Y.K., J.L., D.Y.P., K.S.S., H.C., M.H.R. and M.K. processed clinical samples and conducted clinicopathological correlation analyses. S.U.H. provided samples and supervised this study. H.K.K. designed and supervised this study and wrote a manuscript.

## Additional information

**Competing interests:** The authors declare no competing interests.

