## [Peer Review File · Nature Communications]

Reviewers' Comments:

Reviewer #1:

Remarks to the Author:

The proposed study uses RNA-Seq from 80 patient tumors collected from young (≤ 45 years) patients with diffuse gastric cancer (DGC) to discover gene fusions. Overall, they reconfirm previously known gene fusions in gastric cancer (albeit at slightly different frequencies in younger patients) with one novel, low frequency, gene fusion ANXA2-MYO9A. As such, this findings from this study are likely limited in scope and are a modest advance to what has already been discovered that would warrant publication in a more focused journal.

(1) The authors indicate that "We also performed RNA sequencing on 65 samples of normal tissue adjacent to the tumors.". They should clarify whether these adjacent to the 80 tumor samples that had RNA-Seq data or adjacent to tumors from an independent set of patients.

(2) More specificity should be provided regarding their bioinformatics workflow. In the manuscript they indicated "We applied bioinformatics algorithms such as PRADA or Trans-ABYSS to the RNA-sequencing dataset to predict novel in-frame fusions." This implies that they use PRADA or Trans-ABYSS, but not both? Yet in the methods they also mention using deFuse, FusionMap, and TopHat-Fusion to predict candidate fusion breakpoints. It would be worthwhile to know how many fusions were nominated and filtered at each step. Additionally, how candidates were filtered. For instance, did deFuse, FusionMap, and TopHat-Fusion have to reconfirm the fusion breakpoint?

(3) Given the biological importance of the prioritized fusion, a more comprehensive validation strategy would have been to design probes for the relevant fusion partners (Table 1) to perform a capture strategy coupled with sequencing. This would validate the relevant candidate in the index case while enabling the discovery of additional fusions (even with different fusion partners) in additional samples. This would be very valuable for determining if the fusions are actually occurring at a higher frequency than the authors currently observe. This might also account for the fact that ARHGAP26, RHOA, SLC1A2, UBR4, PGAP3, UBE2L3, and PREX1 were found in the discovery cohort, are known to have different gene fusion partners in the TCGA cohort, but only CLDN18-ARHGAP26, CTNND1-ARHGAP26, and ANXA2-MYO9A were found in 2 or more patients in the 384 validation patients.

Reviewer #2:

Remarks to the Author:

This is a very interesting piece of work and adds to the growing body of knowledge regarding Diffuse Gastric Cancer (DGC) which is increasingly being considered separate to Intestinal type GC. While many DGC are genomically stable (as defined by TCGA) there are an important subgroup that have fusions and involve the RhoA pathway.

I have some queries with respect to the data:

1. Fig 3b, changes in diameter are not impressive (especially for Anxa2-Myo9a) despite statistical difference, what happens to growth curves rather than single timepoints? Also, migration or invasion assays would be more correlative with metastatic and invasive potential and more indicative of DGC in vivo.

2. Fig2g I cannot convincingly say this is a DGC, it seems to be trying to form glands. There is background staining of the normal (these are interstitial cells but unclear as to origin). It would be useful to see a ANXA2-MYO9A negative and positive tumour in terms of IHC.

3. Are any of the fusions co-segregating with ethnicity?

4. Some of the findings were not novel but reproduced other studies, such as mutual exclusivity of

CLDN18-ARHGAP26, CDH1 and RHOA. However, addition of the other fusions adds some novelty.

5. Was there any correlation with *H. pylori* in this cohort. This is usually associated with Intestinal type Gastric Cancer but there still remains an association with DGC. Is there any association with particular fusions or early onset disease that may speak of immunological mechanisms that may play a role in pathogenesis.

Reviewer #3:

Remarks to the Author:

Yang et al carried out analysis of RNA-seq data from early onset diffuse gastric cancer (DGC) to find fusion proteins that may play roles in development of gastric cancer. Authors uncovered 25 in-frame fusion mRNAs and further validation with 384 DGC tumors showed that three fusion are recurrent. All recurrent fusion proteins contain RhoGAP domain. Authors also demonstrated clinical significance of fusion proteins in DGC: patients (n = 17) with these fusion proteins have poor prognosis compared with those without fusion proteins.

Overall, study showed interesting data, especially prognostic association of fusion proteins is novel finding. However, molecular insight on this clinical relevance is lacking.

Major Concerns

1. It will be interesting to see how fusion proteins are prognostically related to CDH1 mutation in DGC: KM plots for Fusion vs. CDH1 mutant vs. CDH1 WT.
2. Functional roles of CLDN18-ARHGAP26 in gastric cancer are not fully characterized. Invasiveness of CLDN18-ARHGAP26 is already demonstrated in previously study as mentioned by authors.
3. Genomic data should be available to readers. All data need to be submitted to GEO before publication.

Minor Concerns

1. In figure 2c, title of y-axis is missing.
2. In figure 4a, gender information appears to be incorrect.

The revised manuscript contains the functional data of *CLDN18-ARHGAP26* as suggested by Reviewer #3 (Fig 5, Supplementary Figs. 10-17, and Supplementary Table 12). We have also added more information (Supplementary Tables 8 and 13), which we believe improves the quality of our work, to our original manuscript, although these data were not requested by Reviewers.

We appreciate reviewers for their positive comments and for comments. Our responses are listed point-by-point below:

Reviewer #1 (Remarks to the Author):

The proposed study uses RNA-Seq from 80 patient tumors collected from young (≤ 45 years) patients with diffuse gastric cancer (DGC) to discover gene fusions. Overall, they reconfirm previously known gene fusions in gastric cancer (albeit at slightly different frequencies in younger patients) with one novel, low frequency, gene fusion ANXA2-MYO9A. As such, these findings from this study are likely limited in scope and are a modest advance to what has already been discovered that would warrant publication in a more focused journal.

(1) The authors indicate that "We also performed RNA sequencing on 65 samples of normal tissue adjacent to the tumors.". They should clarify whether these adjacent to the 80 tumor samples that had RNA-Seq data or adjacent to tumors from an independent set of patients.

RESPONSE: We thank a reviewer for this point. In the revised RESULTS, we have described that We also performed RNA sequencing on 65 samples of normal tissue adjacent to the 80 tumors that had RNA sequencing data (*page 3 line 11*). We also described it in the revised METHODS (*page 8 line 32*).

(2) More specificity should be provided regarding their bioinformatics workflow. In the manuscript they indicated "We applied bioinformatics algorithms such as PRADA or Trans-ABYSS to the RNA-sequencing dataset to predict novel in-frame fusions." This implies that they use PRADA or Trans-ABYSS, but not both? Yet in the methods they also mention using deFuse, FusionMap, and TopHat-Fusion to predict candidate fusion breakpoints. It would be worthwhile to know how many fusions were nominated and filtered at each step. Additionally, how candidates were filtered. For instance, did deFuse, FusionMap, and TopHat-Fusion have to reconstruct the fusion breakpoint?

RESPONSE: This point of a reviewer is well taken. Our original description in RESULTS was confusing (We applied bioinformatics algorithms such as PRADA **or** Trans-ABYSS to the RNA-sequencing dataset to predict novel in-frame fusions, and performed RT-PCR to validate the expression of these in-frame fusion candidates).

We have revised our original description to

We applied bioinformatics algorithms such as PRADA **and** Trans-ABYSS to the RNA-sequencing dataset to predict novel in-frame fusions, and performed RT-PCR to validate the expression of these in-frame fusion candidates (*page 3 line 13*).

Also, we have revised our METHODS as follows:

(page 9 line 35).

To identify gene fusions in DGC, we used RnAseq Data Analysis (PRADA)^{41,42}. In parallel, we used deFuse⁴³, FusionMap⁴⁴, TopHat-Fusion⁴⁵, and Trans-ABYSS⁴⁶ (Supplementary Fig. 18). Using the 'preprocess' module of PRADA, we aligned RNA-Seq reads on human reference genome hg19/GRCh37 and human transcripts of Ensembl build 64. We then obtained gene fusion candidates using the 'fusion' module with options of "-mm 1 -junL 80 -minmapq 30". We selected in-frame gene fusion candidates using the 'prada-frame' utility. RT-PCR was conducted for candidate in-frame fusions.

Additionally, deFuse⁴³, FusionMap⁴⁴, and TopHat-Fusion⁴⁵ were used to predict candidate fusion breakpoints. To remove false-positive breakpoints resulting from these algorithms, we reconstructed candidate regions containing putative breakpoint using Trans-ABYSS^{46,47} (v1.4.4). Trans-ABYSS performed a *de novo* assembly on candidate regions containing putative breakpoints that were identified by deFuse, FusionMap and TopHat-Fusion. We then performed RT-PCR sequencing analyses to validate the expression of candidate fusions that were in-frame and contained partner genes of importance based on the data in the literature (Supplementary Fig. 18).

A full list of candidate fusions that were tested by RT-PCR is shown in Supplementary Table 4.

In Supplementary Fig 18 and Supplementary Methods (page 2 lines 5-32), we have described how many fusions were nominated and filtered at each step.

(3) Given the biological importance of the prioritized fusion, a more comprehensive validation strategy would have been to design probes for the relevant fusion partners (Table 1) to perform a capture strategy coupled with sequencing. This would validate the relevant candidate in the index case while enabling the discovery of additional fusions (even with different fusion partners) in additional samples. This would be very valuable for determining if the fusions are actually occurring at a higher frequency than the authors currently observe. This might also account for the fact that ARHGAP26, RHOA, SLC1A2, UBR4, PGAP3, UBE2L3, and PREX1 were found in the discovery cohort, are known to have different gene fusion partners in the TCGA cohort, but only CLDN18-ARHGAP26, CTNND1-ARHGAP26, and ANXA2-MYO9A were found in 2 or more patients in the 384 validation patients.

RESPONSE: We thank a reviewer for helpful comments. We have added the following paragraph to our DISCUSSION:

This study has a limitation that we used a conventional RT-PCR sequencing method to determine the frequency of gene fusions in the expanded dataset. A hybrid capture-based, targeted RNA sequencing approach, which is more sensitive for fusion detection³⁵, could be used to evaluate if the fusions were actually present at a higher frequency than that observed. This approach was only used to validate the presence of the *EML4-ALK* fusion in a tumor in the discovery set (Supplementary Table 14). (page 8 line 16).

Supplementary Table 14 of the revised manuscript presents the targeted RNA sequencing data of a tumor identified to express *EML4-ALK* by RNA sequencing analysis. Targeted RNA sequencing validated the expression of *EML4-ALK* in this tumor, as both algorithms (ChimeraScan and deFuse) nominated the *EML4-ALK* gene fusion in this sample.

Reviewer #2 (Remarks to the Author):

This is a very interesting piece of work and adds to the growing body of knowledge regarding Diffuse Gastric Cancer (DGC) which is increasingly being considered separate to Intestinal type GC. While many DGC are genomically stable (as defined by TCGA) there are an important subgroup that have fusions and involve the RhoA pathway.

I have some queries with respect to the data:

1. Fig 3b, changes in diameter are not impressive (especially for Anxa2-Myo9a) despite statistical difference, what happens to growth curves rather than single timepoints? Also, migration or invasion assays would be more correlative with metastatic and invasive potential and more indicative of DGC in vivo.

RESPONSE: We appreciate a reviewer for these comments. Growth curves could not be extrapolated from the cell aggregation experiments. We found that the cells aggregated 2 h after setup, and there were no further increases in the size of the aggregate after that time point. As suggested by a reviewer, we have added migration and invasion data to RESULTS (Fig 5) of the revised manuscript.

2. Fig2g I cannot convincingly say this is a DGC, it seems to be trying to form glands. There is background staining of the normal (these are interstitial cells but unclear as to origin). It would be useful to see a ANXA2-MYO9A negative and positive tumour in terms of IHC.

RESPONSE: This point of a reviewer is extremely well taken. The case contained areas of mixed histology. The diffuse histology portion has been evaluated for MYO9A expression and presented in the revised manuscript. There was another DGC case harboring the *ANXA2-MYO9A* fusion, and the MYO9A immunostaining of the case has also been presented in the revised manuscript. As a reviewer suggested, these two DGCs expressing *ANXA2-MYO9A* have been compared for immunostaining with randomly-selected fusion-negative cases in the revised manuscript. MYO9A immunostaining was significantly different between these two groups ($P=0.013$, Cochran-Mantel-Haenszel; Supplementary Table 9). In the revised manuscript, we have presented representative photos for these two groups of DGCs— *i.e.*, those with or without *ANXA2-MYO9A*.

3. Are any of the fusions co-segregating with ethnicity?

RESPONSE: All our tumors were collected from Koreans so that we were not able to address the issue of ethnic difference in the prevalence of RhoGAP domain-containing fusions.

4. Some of the findings were not novel but reproduced other studies, such as mutual exclusivity of CLDN18-ARHGAP26, CDH1 and RHOA. However, addition of the other fusions adds some novelty.

5. Was there any correlation with *H. pylori* in this cohort. This is usually associated with Intestinal type Gastric Cancer but there still remains an association with DGC. Is there any association with particular fusions or early onset disease that may speak of immunological mechanisms that may play a role in pathogenesis.

RESPONSE: This point of a reviewer is well taken. In the revised manuscript, Supplementary Table 6 presents the difference in the prevalence of the *CLDN18-ARHGAP26* fusion between *H. pylori*-positive tumors and *H. pylori*-negative tumors. We also added the following to DISCUSSION (*page 8 line 11*)

We also observed a trend for the relatively high prevalence of the *CLDN18-ARHGAP26* fusion among *H. pylori*-positive DGCs. The association between *H. pylori* and chromosomal translocation has not been reported in gastric adenocarcinomas, unlike in MALT lymphomas. Further studies are warranted to validate this interesting finding

Reviewer #3 (Remarks to the Author):

Yang et al carried out analysis of RNA-seq data from early onset diffuse gastric cancer (DGC) to find fusion proteins that may play roles in development of gastric cancer. Authors uncovered 25 in-frame fusion mRNAs and further validation with 384 DGC tumors showed that three fusion are recurrent. All recurrent fusion proteins contain RhoGAP domain. Authors also demonstrated clinical significance of fusion proteins in DGC: patients (n = 17) with these fusion proteins have poor prognosis compared with those without fusion proteins.

Overall, study showed interesting data, especially prognostic association of fusion proteins is novel finding. However, molecular insight on this clinical relevance is lacking.

Major Concerns

1. It will be interesting to see how fusion proteins are prognostically related to CDH1 mutation in DGC: KM plots for Fusion vs. CDH1 mutant vs. CDH1 WT.

RESPONSE: This point of a reviewer is well taken. We have added Supplementary Figure 9 that

shows KM plots for fusion-positive tumors vs. CDH1-mutated tumors vs. tumors wild-type for both genomic alterations.

In Supplementary Tables 10 and 11 of the revised manuscript, prognostic effects of gene fusions were compared with that of the CDH1 mutation.

Variable	P ¹	HR ² (95% CI)
RhoGAP domain fusion	0.001	3.1 (1.5-6.1)
CDH1 mutation	0.008	2.0 (1.2-3.3)

2. Functional roles of CLDN18-ARHGAP26 in gastric cancer are not fully characterized. Invasiveness of CLDN18-ARHGAP26 is already demonstrated in previously study as mentioned by authors.

RESPONSE: We thank a reviewer for pointing out this important issue. We conducted migration/invasion assays using gastric cancer cell lines and presented the data in Fig 5 of the revised manuscript.

Briefly, mouse gastric cancer cells stably *CLDN18-ARHGAP26* demonstrated significantly enhanced migration ability compared with those expressing an empty vector ($P < 0.001$, Wilcoxon signed rank), and this effect was prominent in *NCC-S1M* and *Pdx1-cre; Smad4^{F/F}; Trp53^{F/F}; Cdh1^{F/+}* cells (Fig 5a). Also, human gastric cancer cells (*NUGC4*, *SNU-719* and *SNU-638*) stably *CLDN18-ARHGAP26* demonstrated significantly enhanced migration capacity compared with those expressing an empty vector ($P < 0.001$, Wilcoxon signed rank). According to the invasion assay, the same set of three human gastric cancer cells stably *CLDN18-ARHGAP26* demonstrated significantly enhanced invasion

capacity compared with those expressing an empty vector ($P < 0.001$, Wilcoxon signed rank).

3. Genomic data should be available to readers. All data need to be submitted to GEO before publication.

RESPONSE: We have submitted the RNA sequencing data to GEO (accession number, GSE110875).

As stated in the original manuscript, we have deposited all of RNA sequencing, whole genome sequencing data, and SNP6.0 array data to the European Genome-phenome Archive database under accession numbers EGAD00001002187, EGAD00010000889, and EGAD00001003953. We have stated the accession numbers in the METHOD section of the revised manuscript.

Minor Concerns

1. In figure 2c, title of y-axis is missing.

RESPONSE: We thank a reviewer for pointing this out. In the revised manuscript, Fig 2c has the title of y-axis.

2. In figure 4a, gender information appears to be incorrect.

RESPONSE: We have corrected gender information in the Fig 4a of the revised manuscript.

Reviewers' Comments:

Reviewer #1:

Remarks to the Author:

The authors have addressed the two technical clarifications. However, the authors have not responded to the most significant concern that most of the findings were not novel (as also pointed out by Reviewer #2 [comment 4]) with the exception of one novel, low frequency, gene fusion ANXA2-MYO9A. As originally suggested, this potentially could have been improved by performing a targeted validation experiment that may reveal additional fusion partners (but similar phenotypic consequences) and thereby increasing the frequency / relevance of some low frequency fusion events. However, since this was not addressed, the overall recommendation remains the same. The findings from this study are a modest advance to what has already been discovered and therefore warrants publication in a more focused journal.

Reviewer #2:

Remarks to the Author:

The authors have addressed my points satisfactorily and I feel have strengthened the manuscript with their revisions.

I have no further comments.

Reviewer #3:

Remarks to the Author:

I have no further concerns.

Reviewers' comments:

Reviewer #1 (Remarks to the Author):

The authors have addressed the two technical clarifications. However, the authors have not responded to the most significant concern that most of the findings were not novel (as also pointed out by Reviewer #2 [comment 4]) with the exception of one novel, low frequency, gene fusion ANXA2-MYO9A. As originally suggested, this potentially could have been improved by performing a targeted validation experiment that may reveal additional fusion partners (but similar phenotypic consequences) and thereby increasing the frequency / relevance of some low frequency fusion events. However, since this was not addressed, the overall recommendation remains the same. The findings from this study are a modest advance to what has already been discovered and therefore warrants publication in a more focused journal.

> Our revised manuscript now presents targeted RNA sequencing data requested by Reviewer #1. During our extensive targeted RNA sequencing analyses, we took extreme care to identify additional novel, recurrent in-frame fusions and to exclude less clinically relevant fusion transcripts such as read-throughs, by conducting parallel, rigorous validation RT-PCR analyses for tumors and adjacent normal tissue. Based on this expanded analysis of 225 qualified diffuse gastric cancers (DGCs), we identified 18 associated novel in-frame fusion genes, including additional recurrent fusion, *TACC2-PPAPDC1A*. All fusion transcripts identified in the three tumors expressing contained *TACC2-PPAPDC1A* contained the PAP2 (type 2 phosphatidic acid phosphatase) domain. We have provided evidence for *PPAPDC1A* mRNA overexpression and relatively high PPAPDC1A (phospholipid phosphatase) activity in the early-onset DGC tumor harboring *TACC2-PPAPDC1A*. We have identified an additional inter-chromosomal *PPAPDC1A* gene fusion event, *ARIH1-PPAPDC1A*. All *PPAPDC1A* fusions were somatic alterations. To our knowledge, we are the first to determine that *TACC2-PPAPDC1A* fusions are recurrent in human cancer.

More importantly, these *PPAPDC1A* fusions were associated with extremely poor prognosis in DGC patients, and their prognostic impact was greater than, and independent of, those of chromosomal instability, which was determined by SNP6.0 array data, and *CDH1* mutations. To our knowledge, similar comprehensive multi-platform genomic studies have not yet been conducted on a larger set of DGCs with long-term follow up (median survival of those without the fusion was > 7 years). Hence, we believe that our data are highly clinically relevant, with regards to the prognostication and the future development of novel therapeutics including personalized immunotherapy for DGC. Our findings suggest novel roles of PAP2 and RhoGAP domains in the progression of DGCs, and possibly of other tumors as well, given that RhoGAP domain-containing fusions and *PPAPDC1A* fusions were recurrent in DGCs and associated with poor prognosis independently from aneuploidy and *CDH1* mutations.

Despite time constraints related to completing revisions, we have completed targeted RNA analyses of 74% of DGCs without available RNA sequencing data, as requested by Reviewer #1, and conducted additional focused experiments for the *TACC2-PPAPDC1A* fusion. We have changed the

manuscript title accordingly and have added these data to the revised RESULTS section, Tables (2 and 3), and Figures (6 and 7).

Reviewer #2 (Remarks to the Author):

The authors have addressed my points satisfactorily and I feel have strengthened the manuscript with their revisions.

I have no further comments.

Reviewer #3 (Remarks to the Author):

I have no further concerns

We believe that we have sufficiently addressed the reviewers' comments and that a revised manuscript is stronger because of their feedback. We look forward to your favorable response regarding our resubmission.

Reviewers' Comments:

Reviewer #1:

Remarks to the Author:

The authors have sufficiently addressed my concerns.

The referee's comment is here;

REVIEWERS' COMMENTS:

Reviewer #1 (Remarks to the Author):

The authors have sufficiently addressed my concerns.

> We are pleased to be able to answer your concerns.